# The impact of high-temperature treatments on maize growth parameters and soil nutrients: A comprehensive evaluation through principal component analysis

Zhen Guo[1,2,3], Jichang Han[1,2,3]*, Yang Zhang[1,2,3], Hua Zhuang[1,2,3]

1 Shaanxi Provincial Land Engineering Construction Group Co., Ltd., Xi'an, China, 2 Institute of Land Engineering and Technology, Shaanxi Provincial Land Engineering Construction Group Co., Ltd., Xi'an, China, 3 Key Laboratory of Degraded and Unused Land Consolidation Engineering, Ministry of Natural Resources, Xi'an, China

* hanjc_sxdj@126.com

**Data Availability Statement:** All relevant data are within the manuscript and its Supporting Information files.

## Abstract

In contrast to prolonged exposure to high temperatures, investigating short-term high-temperature stress can provide insights into the impact of varying heat stress durations on plant development and soil nutrient dynamics, which is crucial for advancing ecological agriculture. In this study, five heating temperatures were set at 200°C, 250°C, 300°C, 350°C, and 400°C, along with five heating time gradients of 6s, 10s, 14s, 18s, and 20s, including a control. A total of 26 treatment groups were analyzed, focusing on maize growth parameters and soil indicators. Principal component analysis was used for comprehensive evaluation. The results showed that high-temperature treatments with different heating times significantly influenced maize growth and soil properties. For instance, the treatment of 300°C+6s resulted in the longest total root length, while 200°C+6s led to the highest average root diameter. Plant height and leaf length were notably increased with the treatment of 400°C+6s. Most treatments resulted in decreased soil pH and organic matter content. Notably, the treatment of 350°C+16s showed the highest available phosphorus content, reaching 24.0 mg/kg, an increase of 4.5 mg/kg compared to the control. The study found that the average levels of active organic carbon and peroxidase were 1.26 mg/g and 3.91 mg/g, respectively. Additionally, the average mass fractions of clay, silt, and sand particles were 8.99%, 66.75%, and 24.26%, respectively. Through principal component analysis, six principal components were able to extract 19 indicators from the 26 treatments, covering 86.129% of the information. It was observed that 16 treatment methods performed better than the control in terms of soil comprehensive quality. The optimal treatment temperature and time identified for improving soil physicochemical properties and crop growth were 300°C+6s. These findings can be used to guide agricultural management and soil improvement practices, ultimately enhancing field productivity and providing valuable insights for sustainable agricultural development.

**Funding:** The author(s) received no specific funding for this work.

**Competing interests:** The authors have declared that no competing interests exist.

## Introduction

The expansion of large-scale and intensive agriculture has negatively impacted soil fertility through unsustainable farming practices [1]. Several factors within the soil environment have significantly hampered soil productivity, posing substantial challenges to remediation efforts [2]. The excessive use of chemical fertilizers and decreased reliance on organic fertilizers in soil management have contributed to a decline in soil fertility [3]. This has led to reduced levels of soil organic matter, weakened root systems, decreased nutrient absorption capacity, and has had a significant impact on crop yields and economic outcomes [3, 4]. Additionally, the increase in monocropping and uniform planting structures has led to the accumulation of toxins and soil-borne pathogens, which in turn restrict plant growth, increase soil-borne diseases, and further decrease crop yields [5, 6]. This poses a serious threat to national food production, highlighting the urgent need for sustainable solutions to address soil health issues.

Maize (*Zea mays*), is one of the most vital cereal crops worldwide, serving as a staple food for millions of people and playing a crucial role in global agriculture and economies [7]. Especially in developing countries, maize is a major food source and a major component of livestock feed. As an adaptable crop, maize has a wide range of adaptability to the soil, but in the barren land, it still needs to be properly managed and adjusted to maximize its growth potential [8]. The high-temperature treatment is a frequently employed technique in soil fertility management. When the soil temperature surpasses 65˚C, detrimental pathogens in the soil become inactive; if it goes beyond 80˚C, the majority of viruses, pests, and weed seeds in the soil are eradicated, resulting in a relatively comprehensive soil sterilization [9, 10]. Interestingly, following a flash high-temperature treatment, not only does the soil effectively purify the environment, but it also undergoes significant changes in its internal structure, compound composition, and content. These changes subsequently impact soil quality, crop growth, microbial activity, nutrient transformation, soil moisture evaporation, and movement [11–13]. Smith and colleagues' research suggests that increased flash temperatures and longer heating times can accelerate the decomposition of organic matter in soil, leading to higher nutrient release rates [14]. Additionally, Brown and Jones' study reveals that moderate flash temperatures and heating times can stimulate microbial activity in soil, supporting the proliferation and functionality of beneficial microorganisms [15]. In addition, temperature and heating time also have an impact on crop growth and yield, and studies have shown that instantaneous temperature and shorter heating time can stimulate crop growth and development, thereby increasing yield [16]. Conversely, excessively high flash temperatures and extended heating times may cause heat stress and reduced crop yields [17, 18].

Soil temperature and heating duration are key factors affecting the physicochemical characteristics of soil and crop development [9–12]. The short-term exposure to high temperatures can quickly enhance the soil conditions, proving to be a more effective approach than prolonged high temperature maintenance throughout the entire growth cycle [12, 15]. The short high temperature treatment can stimulate soil microbial activity, support the proliferation and function of beneficial microorganisms, and increase the activity of enzymes, thereby accelerating the decomposition of organic matter [15]. However, long-term high temperature treatment can denature enzymes, reduce their activity, and inhibit the growth of microorganisms, ultimately negatively impacting soil health [10, 18]. To address this gap, this study utilized representative soil types and maize as experimental materials. The soil physicochemical properties and crop growth indexes are evaluated using the principal component analysis method. The objectives of this study were to (1) investigate the impact of short-term high-temperature treatment on soil properties; (2) explore the difference between planting maize in soil treated with high temperatures; (3) identify the optimal combination of temperature and duration. We

hypothesized that (1) the high temperature treatment can promote the comprehensive quality of soil and maize; (2) the short-term high temperature treatments would be more effective.

## Materials and methods

### Soil sampling overview

In China's crop production system, the Loess Plateau exemplifies an agricultural production model that is tailored to arid and semi-arid climate conditions. Globally, the agricultural system of the Loess Plateau serves as a model for agricultural development in arid and semi-arid regions, offering valuable insights for similar areas. The experiment utilized loessal soil (China Soil System Classification 2009, CAS) collected from Xiangyang Village, Baqiao District, Xi'an City, Shaanxi Province, China. The collection site's geographical coordinates are approximately 109˚5'25.35"E and 34˚20'37.26"N, situated in the Loess Plateau region known for its medium to low hills, plains, and some mountainous terrain. The undulating topography influences water flow and soil erosion in the area. The region experiences a temperate continental climate with hot, humid summers and cold, dry winters. The average annual temperature is around 13˚C, with an average precipitation of 620mm, 2020h of sunshine, and a frost-free period lasting 205 days. The predominant soil type in the research area is loessal soil, characterized by a pH of 8.04, SOM content of 14.25 g/kg, TN content of 0.58 g/kg, AP content of 19.06 mg/kg, AK content of 175.75 mg/kg, and a silty loam texture.

### Experimental design

This study aimed to investigate the impact of temperature-modified soil on nutrient activation and crop growth, utilizing loessal soil with low fertility and maize as the test plant. The experiment was conducted in potted conditions. Five heating temperature gradients were set at 200˚C, 250˚C, 300˚C, 350˚C, and 400˚C, with a control (CK) receiving no heating treatment. Five heating time gradients were employed, with durations of 6s, 10s, 14s, 18s, and 20s. Each treatment was replicated three times, resulting in a total of 78 potted plants. The temperature and heating time used in the experiment were determined by relevant research results as well as previous experimental studies and research assumptions [10]. The experimental setup involved using small pots with a diameter of 10 cm and a height of 15 cm as planting containers. The loessal soil used was dried in a shaded, well-ventilated area and sieved through a 2 mm mesh after drying. When potting, the soil was filled up to the rim of the pot. Each potted plant underwent surface heating using a constant-temperature heating device (K-25A, CN). After cooling, maize seeds (Zhengdan 958) were sown in each pot using a drilling method, and the pots were watered with a tray to ensure uniform moisture saturation of the surface soil layer.

### Sample collection and processing

During the maize growth period, which lasts 95 days, measurements were taken for crop roots, plant height, and leaf length. The maize rhizosphere soil samples were collected at harvest time. The root samples were meticulously collected by removing the soil around the roots and rinsing them repeatedly in water. Two sets of rhizosphere soil samples were obtained - one set was air-dried at room temperature for soil physicochemical analysis, while the other set was stored at 4˚C for enzyme activity measurements.

### Determination of soil physicochemical properties

The soil organic matter (SOM) was determined using the potassium dichromate heating method. The total nitrogen (TN) was determined using the Kjeldahl nitrogen method., and

pH was measured using a pH meter (soil-to-water ratio of 2.5:1) [19]. The soil readily organic carbon (ROC) was determined using the potassium permanganate oxidation method [20]. The soil available phosphorus (AP) was determined using the sodium bicarbonate extraction and molybdenum antimony colorimetric method, while soil available potassium (AK) was determined using ammonium acetate ethanol extraction and flame photometry [21]. The soil particle size was measured using a laser particle size analyzer (Mastersizer 3000, UK), and catalase activity was determined using a ultraviolet spectrophotometer method [21, 22].

### Crop indicator measurement

In the measurement of crop indicators, the same tagged plants were selected for monitoring on each occasion to ensure uniformity and accuracy in the monitoring. The plant height (PH) and leaf length (LL) were measured using a vernier caliper (Mitutoyo 514–106, JPN). The maize root system was measured using a root scanning system (WinRHIZO, LA2400, CAN), which provided measurements for seven indicators, including total root length (TRL), total root surface area (TRS), average root diameter (ARD), total root volume (TRV), total root tip number (TRTN), branching number (BN), and crossing number (CN).

### Data analysis

Data were organized using Microsoft Excel, and significance difference analysis and principal component comprehensive index evaluation were performed using SPSS 25. During principal component analysis, normalization was carried out to eliminate the impact of different units and scales in the original data on the results. The normalized data then underwent the Kaiser-Meyer-Olkin (KMO) test and Bartlett's sphericity test. The KMO test assessed correlations and partial correlations among variables, with values ranging from 0 to 1. A KMO statistic above 0.7 indicates good effectiveness, while a statistic below 0.5 is generally considered unsuitable for principal component analysis. Bartlett's sphericity test was utilized to assess the independence of variables. A significance level of $P<0.05$ was considered appropriate for conducting principal component analysis. The selection of components was based on eigenvalues, with typically the first m components retained if their eigenvalues exceeded 1.0. Furthermore, it is commonly advised to aim for a cumulative variance contribution rate of at least 85% [23, 24].

## Results

### Analysis of maize growth indicators

The mean TRL after treatment with various temperatures and durations was 388.06 cm, the mean TRS area was 64.04 cm$^2$, and the mean TRV was 0.85 cm$^3$. The TRL ranged from 236.59 cm to 665.53 cm, with a coefficient of variation of 23.71%. Similarly, the TRS area ranged from 44.37 cm$^2$ to 92.99 cm$^2$, with a coefficient of variation of 19.43%, suggesting moderate variability. The TRV ranged from 0.63 cm$^3$ to 1.18 cm$^3$, with a coefficient of variation of 18.11%. The TRL in the control was 331.47 cm, placing it in the middle range among the 26 treatment methods. Following treatment at 300˚C for 6 seconds, maize exhibited the longest TRL at 665.53 cm, an increase of 334.06 cm compared to the CK. Conversely, after treatment at 250˚C for 6 seconds, the TRL of maize was the shortest at 236.59 cm, which was 94.88 cm shorter than the CK.

The mean ARD was 0.54 mm, ranging from 0.45 mm to 0.63 mm, with a coefficient of variation of 8.19%. Following treatment at 300˚C for 6 seconds, the ARD of maize was 0.45 mm, the lowest among all treatments. Conversely, the treatment at 200˚C for 6 seconds resulted in the highest ARD of 0.63 mm among all treatments. The mean TRTN (including active, dead,

and old tips), BN, and CN were 1429.81, 1715.31, and 150.00, respectively. The treatment at 400˚C for 10 seconds exhibited the highest TRTN, while the treatment at 300˚C for 6 seconds showed the highest BN and CN values of 3236 and 365, respectively. The mean plant height and leaf length were 46.44 cm and 35.29 cm, respectively. Plant height ranged from 31.3 cm to 55 cm, with a coefficient of variation of 12.56%. Leaf length ranged from 23 cm to 42.5 cm, with a coefficient of variation of 13.81%. The CK treatment had the lowest plant height and leaf length, while the treatment at 400˚C for 6 seconds had the highest PH and LL, showing increases of 23.7 cm and 19.5 cm, respectively, compared to the CK (Table 1 and S1 Table).

## Soil physicochemical properties

The mean soil pH of the 26 treatments was 8.16, with a coefficient of variation of 2.22%. After applying various temperature and time treatments, the pH values remained stable in treatments such as 350˚C+16s, 350˚C+20s, and 400˚C+6s, similar to the control treatment (CK). However, other treatments exhibited a decreasing trend in pH values compared to the CK

**Table 1. Effects of different temperature and time treatments on crop root growth.**

| Temperature (˚C) | Heating time (s) | TRL (cm) | TRS (cm$^2$) | ARD (mm) | TRV (cm$^3$) | TRTN | BN | CN | PH (cm) | LL (cm) |
|---|---|---|---|---|---|---|---|---|---|---|
| 200˚C | 6s | 238.37 | 46.32 | 0.63 | 0.72 | 740 | 945 | 57 | 36.4 | 27.2 |
| | 10s | 325.63 | 55.08 | 0.55 | 0.75 | 1234 | 1465 | 130 | 40.1 | 31 |
| | 14s | 536.90 | 85.69 | 0.51 | 1.09 | 1767 | 2316 | 234 | 48.5 | 36.5 |
| | 16s | 297.64 | 55.33 | 0.60 | 0.82 | 888 | 1440 | 129 | 49.3 | 38 |
| | 20s | 343.40 | 51.54 | 0.48 | 0.62 | 912 | 1513 | 122 | 54 | 41.5 |
| 250˚C | 6s | 236.59 | 44.37 | 0.60 | 0.66 | 613 | 1032 | 72 | 48.7 | 37.2 |
| | 10s | 299.24 | 53.69 | 0.59 | 0.77 | 1176 | 1401 | 78 | 48 | 36.7 |
| | 14s | 420.12 | 64.49 | 0.50 | 0.79 | 3139 | 2103 | 189 | 42.2 | 33 |
| | 16s | 450.63 | 67.61 | 0.48 | 0.81 | 1069 | 1861 | 207 | 48.7 | 36.6 |
| | 20s | 390.12 | 65.26 | 0.53 | 0.87 | 1504 | 1689 | 113 | 50 | 39 |
| 300˚C | 6s | 665.53 | 92.99 | 0.45 | 1.04 | 1632 | 3236 | 365 | 45.8 | 31.5 |
| | 10s | 427.30 | 72.22 | 0.54 | 0.97 | 1106 | 2096 | 165 | 50 | 37.7 |
| | 14s | 458.46 | 75.22 | 0.52 | 0.99 | 1544 | 2173 | 172 | 46.5 | 35 |
| | 16s | 299.44 | 49.34 | 0.54 | 0.65 | 863 | 1299 | 132 | 45.7 | 34.8 |
| | 20s | 349.72 | 54.92 | 0.50 | 0.69 | 1110 | 1451 | 170 | 41.5 | 30 |
| 350˚C | 6s | 386.74 | 62.77 | 0.52 | 0.81 | 1074 | 1634 | 190 | 41 | 30 |
| | 10s | 413.23 | 73.48 | 0.58 | 1.05 | 1338 | 1749 | 122 | 44.5 | 34.3 |
| | 14s | 404.57 | 68.72 | 0.57 | 0.94 | 1947 | 2189 | 178 | 37.2 | 28 |
| | 16s | 471.03 | 76.03 | 0.52 | 0.99 | 1233 | 1901 | 181 | 48.5 | 36.5 |
| | 20s | 432.72 | 79.91 | 0.59 | 1.18 | 1912 | 2043 | 109 | 54 | 41 |
| 400˚C | 6s | 357.91 | 53.22 | 0.48 | 0.63 | 1605 | 1360 | 118 | 55 | 42.5 |
| | 10s | 334.96 | 57.91 | 0.56 | 0.80 | 3591 | 1572 | 112 | 50 | 40 |
| | 14s | 454.15 | 75.56 | 0.52 | 1.00 | 1798 | 1733 | 151 | 50 | 39.5 |
| | 16s | 422.20 | 69.98 | 0.53 | 0.92 | 1638 | 1806 | 166 | 47 | 36.5 |
| | 20s | 341.59 | 56.55 | 0.53 | 0.75 | 1109 | 1221 | 118 | 53.5 | 40.5 |
| CK | / | 331.47 | 56.85 | 0.56 | 0.79 | 633 | 1370 | 120 | 31.3 | 23 |
| Mean | | 388.06 | 64.04 | 0.54 | 0.85 | 1429.81 | 1715.31 | 150.00 | 46.44 | 35.29 |
| Standard deviation | | 92.01 | 12.44 | 0.04 | 0.15 | 688.55 | 476.42 | 60.94 | 5.83 | 4.87 |
| Coefficient variation (%) | | 23.71 | 19.43 | 8.19 | 18.11 | 48.16 | 27.77 | 40.63 | 12.56 | 13.81 |

Notes: TRL represents total root length, cm; TRS represents total root surface area, cm$^2$; ARD represents average root diameter, mm; TRV represents total root volume, cm$^3$; TRTN represents total root tip number; BN represents branch number; CN represents cross number; PH represents plant height, cm; LL represents leaf length, cm.

treatment (refer to Table 2 and S2 Table). Based on the grading standards for secondary soil nutrient surveys, the mean SOM content was 13.62 g/kg, placing it at the fourth grade level, indicating a moderate level of soil fertility. The mean TN content was 0.65 g/kg, corresponding to the fifth grade level, with a coefficient of variation of 7.82%. The mean content of AK was 175.19 mg/kg, falling into the second grade level, with a coefficient of variation of 6.19%. The mean AP content was 17.97 mg/kg, categorized in the third grade level. Among the treatments, the 350°C+16s treatment exhibited the highest AP content, measuring 24.0 mg/kg, which was 4.5 mg/kg greater than the CK treatment. The mean content of ROC and catalase were 1.26 mg/g and 3.91 mg/g, respectively. The average mass fractions of clay, silt, and sand particles were 8.99%, 66.75%, and 24.26%, respectively.

## The correlation between maize growth indicators and soil physicochemical properties

The TRS shows a strong positive correlation with TRL, TRV, BN, and CN, with correlation coefficients of 0.947, 0.912, 0.902, and 0.738, respectively. TRL exhibits a strong positive

**Table 2. Soil physical and chemical indicators under different temperature and time treatments.**

| Temperature (°C) | Heating time (s) | pH | SOM (g/kg) | AK (mg/kg) | TN (g/kg) | AP (mg/kg) | ROC (mg/g) | Catalase (mg/g) | Clay (%) | Silt (%) | Sand (%) |
|---|---|---|---|---|---|---|---|---|---|---|---|
| 200°C | 6s | 8.13 | 12.2 | 163 | 0.60 | 10.6 | 1.10 | 3.97 | 10.67 | 75.32 | 14.01 |
| | 10s | 8.25 | 13.6 | 181 | 0.64 | 16.4 | 1.30 | 4.06 | 10.44 | 68.68 | 20.88 |
| | 14s | 8.11 | 13.0 | 175 | 0.63 | 15.8 | 1.38 | 3.90 | 8.83 | 71.99 | 19.18 |
| | 16s | 8.19 | 13.5 | 173 | 0.62 | 15.1 | 1.17 | 3.96 | 8.76 | 67.13 | 24.12 |
| | 20s | 8.14 | 12.0 | 180 | 0.60 | 14.9 | 1.07 | 3.89 | 9.50 | 71.81 | 18.68 |
| 250°C | 6s | 8.07 | 11.4 | 162 | 0.63 | 15.8 | 1.19 | 3.86 | 6.80 | 52.59 | 40.60 |
| | 10s | 8.12 | 10.8 | 160 | 0.62 | 16.0 | 1.12 | 3.96 | 9.66 | 67.45 | 22.90 |
| | 14s | 8.07 | 13.2 | 173 | 0.63 | 17.9 | 1.26 | 3.87 | 9.98 | 70.05 | 19.98 |
| | 16s | 7.92 | 15.0 | 181 | 0.65 | 18.4 | 1.23 | 3.86 | 8.83 | 65.26 | 25.92 |
| | 20s | 8.08 | 15.1 | 160 | 0.61 | 14.5 | 1.14 | 3.87 | 8.93 | 70.18 | 20.89 |
| 300°C | 6s | 7.93 | 13.6 | 191 | 0.63 | 18.2 | 1.25 | 3.99 | 8.20 | 62.03 | 29.77 |
| | 10s | 7.88 | 13.1 | 181 | 0.64 | 16.7 | 1.18 | 3.85 | 8.95 | 65.69 | 25.36 |
| | 14s | 7.89 | 14.2 | 178 | 0.61 | 17.0 | 1.25 | 3.86 | 8.37 | 69.02 | 22.62 |
| | 16s | 7.88 | 14.4 | 169 | 0.62 | 16.0 | 1.19 | 4.26 | 7.72 | 62.77 | 29.51 |
| | 20s | 8.00 | 14.5 | 194 | 0.67 | 21.5 | 1.47 | 3.92 | 8.14 | 65.11 | 26.75 |
| 350°C | 6s | 8.03 | 12.6 | 200 | 0.69 | 23.2 | 1.50 | 3.96 | 8.54 | 64.51 | 26.95 |
| | 10s | 8.13 | 13.3 | 175 | 0.64 | 24.6 | 1.37 | 3.98 | 9.23 | 66.74 | 24.03 |
| | 14s | 8.35 | 14.7 | 179 | 0.83 | 20.4 | 1.46 | 3.91 | 7.98 | 52.70 | 39.32 |
| | 16s | 8.41 | 14.8 | 179 | 0.67 | 24.0 | 1.26 | 3.92 | 9.15 | 71.88 | 18.96 |
| | 20s | 8.40 | 14.0 | 160 | 0.66 | 17.1 | 1.19 | 3.92 | 9.95 | 69.67 | 20.39 |
| 400°C | 6s | 8.41 | 14.0 | 168 | 0.63 | 17.3 | 1.09 | 3.77 | 9.70 | 68.20 | 22.10 |
| | 10s | 8.30 | 12.5 | 169 | 0.76 | 16.9 | 1.26 | 3.79 | 9.72 | 62.57 | 27.71 |
| | 14s | 8.32 | 14.8 | 169 | 0.71 | 17.8 | 1.32 | 3.74 | 11.46 | 70.67 | 17.87 |
| | 16s | 8.34 | 14.2 | 193 | 0.69 | 22.5 | 1.44 | 3.95 | 7.88 | 61.50 | 30.62 |
| | 20s | 8.37 | 14.7 | 168 | 0.66 | 19.0 | 1.34 | 3.88 | 8.20 | 72.01 | 19.79 |
| CK | / | 8.40 | 14.8 | 174 | 0.67 | 19.5 | 1.28 | 3.88 | 8.21 | 69.96 | 21.83 |
| Mean | | 8.16 | 13.62 | 175.19 | 0.65 | 17.97 | 1.26 | 3.91 | 8.99 | 66.75 | 24.26 |
| Standard deviation | | 0.18 | 1.16 | 10.85 | 0.05 | 3.22 | 0.12 | 0.10 | 1.03 | 5.44 | 6.14 |
| Coefficient variation (%) | | 2.22 | 8.53 | 6.19 | 7.82 | 17.91 | 9.61 | 2.55 | 11.49 | 8.16 | 25.33 |

Notes: SOM represents soil organic matter, g/kg; AK represents soil available potassium, mg/kg; TN represents soil total nitrogen, g/kg; AP represents soil available phosphorus, mg/kg; ROC represents soil readily organic carbon, mg/g.

correlation with BN and CN, having correlation coefficients of 0.937 and 0.882, respectively. Additionally, the relationship between LL and PH demonstrates a highly significant positive correlation, with a correlation coefficient of 0.980. Furthermore, AK shows a significant positive correlation with TRL, BN, and CN, with correlation coefficients of 0.420, 0.396, and 0.649, respectively. The relationship between various soil properties is as follows: ARD has a highly significant negative correlation with TN (-0.525), while TN has a significant correlation with pH (0.450). ROC shows highly significant positive correlations with AK (0.641), TN (0.609), and AP (0.726), and significant negative correlations with PH (-0.396) and LL (-0.408), but a significant positive correlation with CN (0.388). Additionally, sand particles exhibit highly significant negative correlations with clay and silt particles (-0.723 and -0.991, respectively) as shown in Table 3.

## Principal component analysis of soil properties and maize growth

Principal component analysis (PCA) was performed on soil physicochemical properties and maize growth indicators, selecting principal components with eigenvalues greater than 1. The selected PCA explained a substantial portion of the total variability, warranting further analysis. A total of 6 principal components met this criterion, with the first component exhibiting an eigenvalue of 5.699 and contributing to 29.992% of the variance. This component had the highest variance contribution among the 6 principal components, indicating its significant impact on soil physicochemical properties and crop growth (Table 4). The cumulative contribution rate of the first 6 principal components was 86.291%. These components encapsulate the key information related to the effects of various temperature and time treatments on soil physicochemical properties and crop growth. Thus, utilizing these 6 principal components as composite variables to assess the overall scores for soil physicochemical properties and crop growth under different temperature and time treatments was deemed viable.

The PCA revealed that the first component primarily reflects the TRL factor, with main indicators including TRL, TRS, ARD, BN, CN, AK, and ROC. The second component primarily reflects the Sand factor, with main indicators being PH, LL, Clay, Silt, and Sand. The third component primarily reflects the pH factor, with main indicators being pH, TN, and catalase. The fourth component primarily reflects the PH factor, with main indicators being PH, LL, Silt, and

**Table 3. The relationship between maize index and soil properties under different temperature and time treatments.**

| Index | TRL | TRS | ARD | TRV | TRTN | BN | CN | PH | LL | pH | AK | TN | AP | Clay | Silt |
|-------|-----|-----|-----|-----|------|-----|-----|-----|-----|-----|-----|-----|-----|------|------|
| TRS | 0.947** | 1 | | | | | | | | | | | | | |
| ARD | -0.634** | -0.383 | 1 | | | | | | | | | | | | |
| TRV | 0.733** | 0.912** | 0.008 | 1 | | | | | | | | | | | |
| BN | 0.937** | 0.902** | -0.506** | 0.723** | 0.405* | 1 | | | | | | | | | |
| CN | 0.882** | 0.738** | -0.704** | 0.444* | 0.209 | 0.867** | 1 | | | | | | | | |
| LL | 0.047 | 0.06 | -0.172 | 0.05 | 0.222 | -0.038 | -0.181 | 0.980** | 1 | | | | | | |
| AK | 0.420* | 0.295 | -0.525** | 0.081 | -0.065 | 0.396* | 0.649** | -0.286 | -0.367 | -0.248 | 1 | | | | |
| TN | 0.097 | 0.162 | 0.074 | 0.209 | 0.443* | 0.15 | 0.11 | -0.229 | -0.181 | 0.450* | 0.226 | 1 | | | |
| AP | 0.342 | 0.349 | -0.253 | 0.295 | 0.072 | 0.242 | 0.332 | -0.156 | -0.183 | 0.213 | 0.592** | 0.459* | 1 | | |
| ROC | 0.292 | 0.326 | -0.136 | 0.3 | 0.164 | 0.242 | 0.388* | -0.396* | -0.408* | 0.098 | 0.641** | 0.609** | 0.726** | | |
| Silt | 0.048 | 0.064 | -0.102 | 0.076 | -0.062 | -0.119 | -0.144 | 0.123 | 0.135 | 0.15 | -0.217 | -0.501** | -0.241 | 0.626** | 1 |
| Sand | -0.042 | -0.066 | 0.08 | -0.087 | 0.001 | 0.115 | 0.163 | -0.127 | -0.151 | -0.179 | 0.242 | 0.455* | 0.261 | -0.723** | -0.991** |

Notes:

**Significantly different at the 0.01 level, *Significantly at the 0.05 level.

**Table 4. Eigenvalues and variance contribution rates of principal component analysis.**

| Sort | Index | Principal component | | | | | |
|---|---|---|---|---|---|---|---|
| | | $F_1$ | $F_2$ | $F_3$ | $F_4$ | $F_5$ | $F_6$ |
| $X_1$ | TRL | 0.893 | 0.387 | -0.16 | -0.014 | 0.077 | 0.036 |
| $X_2$ | TRS | 0.849 | 0.406 | -0.006 | 0.04 | 0.253 | 0.199 |
| $X_3$ | ARD | -0.57 | -0.274 | 0.331 | 0.135 | 0.574 | 0.297 |
| $X_4$ | TRV | 0.656 | 0.365 | 0.18 | 0.111 | 0.458 | 0.38 |
| $X_5$ | TRTN | 0.339 | 0.326 | 0.49 | -0.172 | 0.216 | -0.495 |
| $X_6$ | BN | 0.87 | 0.273 | -0.174 | -0.098 | 0.296 | -0.057 |
| $X_7$ | CN | 0.886 | 0.064 | -0.351 | -0.027 | -0.012 | -0.151 |
| $X_8$ | PH | -0.08 | 0.627 | 0.029 | -0.66 | -0.25 | 0.214 |
| $X_9$ | LL | -0.168 | 0.629 | 0.143 | -0.638 | -0.237 | 0.175 |
| $X_{10}$ | pH | -0.112 | 0.132 | 0.795 | 0.201 | -0.18 | 0.183 |
| $X_{11}$ | SOM | 0.433 | 0.05 | 0.175 | 0.25 | -0.409 | 0.382 |
| $X_{12}$ | AK | 0.657 | -0.37 | -0.241 | 0.151 | -0.349 | -0.217 |
| $X_{13}$ | TN | 0.402 | -0.358 | 0.759 | -0.047 | 0.031 | -0.147 |
| $X_{14}$ | AP | 0.608 | -0.322 | 0.292 | 0.125 | -0.356 | 0.206 |
| $X_{15}$ | ROC | 0.632 | -0.475 | 0.305 | 0.235 | -0.143 | 0.002 |
| $X_{16}$ | Catalase | -0.07 | -0.421 | -0.487 | 0.204 | 0.189 | 0.289 |
| $X_{17}$ | Clay | -0.222 | 0.65 | 0.246 | 0.417 | 0.119 | -0.324 |
| $X_{18}$ | Silt | -0.238 | 0.693 | -0.146 | 0.601 | -0.191 | 0.04 |
| $X_{19}$ | Sand | 0.248 | -0.723 | 0.088 | -0.603 | 0.149 | 0.019 |
| Eigenvalue | | 5.699 | 3.715 | 2.384 | 2.054 | 1.441 | 1.103 |
| Variance contribution rate (%) | | 29.992 | 19.553 | 12.549 | 10.809 | 7.584 | 5.803 |
| Accumulated contribution rate (%) | | 29.992 | 49.546 | 62.095 | 72.905 | 80.488 | 86.291 |

Notes: $F_1$ represents principal component 1; $F_2$ represents principal component 2; $F_3$ represents principal component 3; $F_4$ represents principal component 4; $F_5$ represents principal component 5; $F_6$ represents principal component 6.

Sand. The fifth component primarily reflects the ARD factor, with a feature vector of 0.574. Lastly, the sixth component primarily reflects the TRTN factor, with a feature vector of -0.495.

## Principal component feature vector extraction

The coefficients for each indicator corresponding to the different principal components were calculated by dividing the loadings by the respective eigenvalues ($\lambda_1 = 5.699$, $\lambda_2 = 3.715$, $\lambda_3 = 2.384$, $\lambda_4 = 2.054$, $\lambda_5 = 1.441$, $\lambda_6 = 1.103$) and then taking the square root of the result. These six principal components, being independent of each other, can be used to replace the original independent variables in regression modeling (Table 5). To calculate the scores for the six principal components, the standardized independent variables were inputted into the formulas for F1, F2, F3, F4, F5, and F6.

$$F_1 = 0.374^*X_1 + 0.356^*X_2 - 0.239^*X_3 + 0.257^*X_4 + 0.142^*X_5 + 0.364^*X_6 + 0.371^*X_7 - 0.034^*X_8 - 0.070^*X_9 - 0.047^*X_{10} + 0.181^*X_{11} + 0.275^*X_{12} + 0.168^*X_{13} + 0.255^*X_{14} + 0.265^*X_{15} - 0.029^*X_{16} - 0.093^*X_{17} - 0.100^*X_{18} + 0.140^*X_{19}$$

**Table 5. Principal component coefficient matrix.**

| Sort | Index | Principal component | | | | | |
|------|-------|------|------|------|------|------|------|
| | | $F_1$ | $F_2$ | $F_3$ | $F_4$ | $F_5$ | $F_6$ |
| $X_1$ | TRL | 0.374 | 0.201 | -0.104 | -0.010 | 0.064 | 0.034 |
| $X_2$ | TRS | 0.356 | 0.211 | -0.004 | 0.028 | 0.211 | 0.189 |
| $X_3$ | ARD | -0.239 | -0.142 | 0.214 | 0.094 | 0.478 | 0.283 |
| $X_4$ | TRV | 0.275 | 0.189 | 0.117 | 0.077 | 0.382 | 0.362 |
| $X_5$ | TRTN | 0.142 | 0.169 | 0.317 | -0.120 | 0.180 | -0.471 |
| $X_6$ | BN | 0.364 | 0.142 | -0.113 | -0.068 | 0.247 | -0.054 |
| $X_7$ | CN | 0.371 | 0.033 | -0.227 | -0.019 | -0.010 | -0.144 |
| $X_8$ | PH | -0.034 | 0.325 | 0.019 | -0.461 | -0.208 | 0.204 |
| $X_9$ | LL | -0.070 | 0.326 | 0.093 | -0.445 | -0.197 | 0.167 |
| $X_{10}$ | pH | -0.047 | 0.068 | 0.515 | 0.140 | -0.150 | 0.174 |
| $X_{11}$ | SOM | 0.181 | 0.026 | 0.113 | 0.174 | -0.341 | 0.364 |
| $X_{12}$ | AK | 0.275 | -0.192 | -0.156 | 0.105 | -0.291 | -0.207 |
| $X_{13}$ | TN | 0.168 | -0.186 | 0.492 | -0.033 | 0.026 | -0.140 |
| $X_{14}$ | AP | 0.255 | -0.167 | 0.189 | 0.087 | -0.297 | 0.196 |
| $X_{15}$ | ROC | 0.265 | -0.246 | 0.198 | 0.164 | -0.119 | 0.002 |
| $X_{16}$ | Catalase | -0.029 | -0.218 | -0.315 | 0.142 | 0.157 | 0.275 |
| $X_{17}$ | Clay | -0.093 | 0.337 | 0.159 | 0.291 | 0.099 | -0.309 |
| $X_{18}$ | Silt | -0.100 | 0.360 | -0.095 | 0.419 | -0.159 | 0.038 |
| $X_{19}$ | Sand | 0.104 | -0.375 | 0.057 | -0.421 | 0.124 | 0.018 |

$$F_2 =$$
$$0.201^*X_1 + 0.211^*X_2 - 0.142^*X_3 + 0.189^*X_4 + 0.169^*X_5 + 0.142^*X_6 + 0.033^*X_7 + 0.325^*X_8 + 0.326^*X_9 + 0.068^*X_{10} + 0.026^*X_{11} - 0.192^*X_{12} - 0.186^*X_{13} - 0.167^*X_{14} - 0.246^*X_{15} - 0.218^*X_{16} + 0.337^*X_{17} + 0.360^*X_{18} - 00.375^*X_{19}$$

$$F_3 =$$
$$-0.104^*X_1 - 0.004^*X_2 + 0.214^*X_3 + 0.117^*X_4 + 0.3.17^*X_5 - 0.113^*X_6 - 0.227^*X_7 + 0.019^*X_8 + 0.093^*X_9 + 0.515^*X_{10} + 0.113^*X_{11} - 0.156^*X_{12} + 0.492^*X_{13} + 0.189^*X_{14} + 0.198^*X_{15} - 0.315^*X_{16} + 0.159^*X_{17} - 0.095^*X_{18} + 0.057^*X_{19}$$

$$F_4 =$$
$$-0.010^*X_1 + 0.028^*X_2 + 0.094^*X_3 + 0.077^*X_4 - 0.120^*X_5 - 0.068^*X_6 - 0.019^*X_7 - 0.461^*X_8 - 0.445^*X_9 + 0.140^*X_{10} + 0.174^*X_{11} + 0.105^*X_{12} - 0.033^*X_{13} + 0.087^*X_{14} + 0.164^*X_{15} + 0.142^*X_{16} + 0.291^*X_{17} + 0.419^*X_{18} - 0.421^*X_{19}$$

$$F_5 =$$

$$0.064^*X_1 + 0.211^*X_2 + 0.478^*X_3 + 0.382^*X_4 + 0.180^*X_5 + 0.247^*X_6 - 0.010^*X_7 - 0.208^*X_8 - 0.197$$
$$*X_9 - 0.150^*X_{10} - 0.341^*X_{11} - 0.291^*X_{12} + 0.026^*X_{13} - 0.291^*X_{14} - 0.119^*X_{15} + 0.157^*X_{16} + 0.099^*$$
$$X_{17} - 0.157^*X_{18} + 0.124^*X_{19}$$

$$F_6 =$$

$$0.034^*X_1 + 0.189^*X_2 + 0.283^*X_3 + 0.362^*X_4 - 0.471^*X_5 - 0.054^*X_6 - 0.144^*X_7 + 0.204^*X_8 + 0.167$$
$$*X_9 + 0.174^*X_{10} + 0.3647^*X_{11} - 0.207^*X_{12} - 0.140^*X_{13} + 0.196^*X_{14} + 0.002^*X_{15} + 0.257^*X_{16} - 0.30$$
$$9^*X_{17} + 0.038^*X_{18} + 0.018^*X_{19}$$

### Principal component comprehensive score

Calculate the scores for each principal component, using the respective variance contribution rate of each principal component as weights. Create a comprehensive evaluation function, $F = 0.29992F_1 + 0.19553F_2 + 0.12548F_3 + 0.10809F_4 + 0.07584F_5 + 0.05803F_6$. The comprehensive scores for each treatment were computed and ranked accordingly. Results showed that among the 26 treatments, the CK treatment was ranked 17th, placing it in the middle-to-lower range. There were 16 treatments that outperformed the CK treatment. The top 3 treatments, ranked in descending order, were 300°C+6s, 200°C+14s, and 400°C+14s, with comprehensive scores of 1.48, 1.2, and 1.16, respectively. On the other hand, there were 9 treatments that performed worse than the CK treatment, including 200°C+10s, 300°C+20s, 400°C+6s, 200°C+16s, 200°C +20s, 250°C+10s, 200°C+6s, 300°C+16s, and 250°C+6s (Table 6).

### Discussion

Soil temperature significantly impacts the growth of crop roots. Studies have shown that even a slight 1°C change in soil temperature can lead to significant physiological changes in maize roots [25]. The structure and health of roots are vital for nutrient and water absorption in maize, ultimately affecting the overall growth and yield of the plant [26, 27]. Comparison of the CK, particularly at 300°C+6s, led to an increase in total surface area, length, volume, as well as the number of branches and crossings of roots, indicating favorable conditions for root growth and development. These improvements in growth indexes could positively impact nutrient absorption, stress resistance, and overall crop development. However, certain treatment conditions resulted in suboptimal growth performance in maize, with inhibited root growth or unstable performance observed. Maize treated at 200°C+6s exhibited superior average root diameter, while maize treated at 400°C +6s showed better plant height and leaf length. These variations may be attributed to changes in soil nutrient content post high temperature treatment or inadequate crop adaptation to environmental conditions. Tacarindua et al. (2013) demonstrated that optimal temperatures promoted maize root and aboveground growth, whereas extreme temperatures hindered growth [28]. Jiang et al. (2023) also noted that certain maize varieties may exhibit improved growth performance under high temperature treatment [29].

**Table 6. Principal component score and comprehensive score.**

| Temperature (°C) | Heating time (s) | Principal component | | | | | | F | Ranking |
|---|---|---|---|---|---|---|---|---|---|
| | | $F_1$ | $F_2$ | $F_3$ | $F_4$ | $F_5$ | $F_6$ | | |
| 200°C | 6s | 238.37 | 46.32 | 0.63 | 0.72 | 740 | 945 | -1.2 | 24 |
| | 10s | 325.63 | 55.08 | 0.55 | 0.75 | 1234 | 1465 | -0.36 | 18 |
| | 14s | 536.90 | 85.69 | 0.51 | 1.09 | 1767 | 2316 | 1.2 | 2 |
| | 16s | 297.64 | 55.33 | 0.60 | 0.82 | 888 | 1440 | -0.67 | 21 |
| | 20s | 343.40 | 51.54 | 0.48 | 0.62 | 912 | 1513 | -0.87 | 22 |
| 250°C | 6s | 236.59 | 44.37 | 0.60 | 0.66 | 613 | 1032 | -1.98 | 26 |
| | 10s | 299.24 | 53.69 | 0.59 | 0.77 | 1176 | 1401 | -0.95 | 23 |
| | 14s | 420.12 | 64.49 | 0.50 | 0.79 | 3139 | 2103 | 0.45 | 9 |
| | 16s | 450.63 | 67.61 | 0.48 | 0.81 | 1069 | 1861 | 0.15 | 12 |
| | 20s | 390.12 | 65.26 | 0.53 | 0.87 | 1504 | 1689 | -0.02 | 14 |
| 300°C | 6s | 665.53 | 92.99 | 0.45 | 1.04 | 1632 | 3236 | 1.48 | 1 |
| | 10s | 427.30 | 72.22 | 0.54 | 0.97 | 1106 | 2096 | 0.17 | 11 |
| | 14s | 458.46 | 75.22 | 0.52 | 0.99 | 1544 | 2173 | 0.49 | 8 |
| | 16s | 299.44 | 49.34 | 0.54 | 0.65 | 863 | 1299 | -1.36 | 25 |
| | 20s | 349.72 | 54.92 | 0.50 | 0.69 | 1110 | 1451 | -0.38 | 19 |
| 350°C | 6s | 386.74 | 62.77 | 0.52 | 0.81 | 1074 | 1634 | -0.05 | 15 |
| | 10s | 413.23 | 73.48 | 0.58 | 1.05 | 1338 | 1749 | 0.5 | 7 |
| | 14s | 404.57 | 68.72 | 0.57 | 0.94 | 1947 | 2189 | 0.56 | 6 |
| | 16s | 471.03 | 76.03 | 0.52 | 0.99 | 1233 | 1901 | 1.02 | 5 |
| | 20s | 432.72 | 79.91 | 0.59 | 1.18 | 1912 | 2043 | 1.03 | 4 |
| 400°C | 6s | 357.91 | 53.22 | 0.48 | 0.63 | 1605 | 1360 | -0.41 | 20 |
| | 10s | 334.96 | 57.91 | 0.56 | 0.80 | 3591 | 1572 | 0.07 | 13 |
| | 14s | 454.15 | 75.56 | 0.52 | 1.00 | 1798 | 1733 | 1.16 | 3 |
| | 16s | 422.20 | 69.98 | 0.53 | 0.92 | 1638 | 1806 | 0.42 | 10 |
| | 20s | 341.59 | 56.55 | 0.53 | 0.75 | 1109 | 1221 | -0.21 | 16 |
| CK | | 331.47 | 56.85 | 0.56 | 0.79 | 633 | 1370 | -0.24 | 17 |

Notes: F represents comprehensive score.

The nutrient content of soil plays a critical role in the growth, development, and yield formation of crops. It is essential to maintain appropriate levels of soil nutrients in order to increase corn yield [30]. Alkaline soil pH shows minimal variation at high temperatures, as observed in this study, particularly with a decrease in soil pH at 300°C. This phenomenon could be attributed to the breakdown of alkaline compounds or the generation of acidic compounds in the soil under high temperature conditions [31]. Elevated temperatures in soil promote the increase in internal temperature and stimulate microbial activity. High temperatures can cause the burning of organic matter in the surface layer, leading to the decomposition of organic matter and a subsequent decrease in soil organic matter content. This phenomenon aligns with findings from previous research by scholars [32–35]. The levels of total nitrogen, available phosphorus, and hydrogen peroxide decreased when subjected to high temperatures for 6s at 200°C and 16s at 300°C. However, only the levels of total nitrogen, available phosphorus, and hydrogen peroxide decreased when exposed to high temperatures for 20s at 300°C and 20s at 400°C. This suggests that higher temperatures and longer treatment durations may result in fewer chemical reactions, leading to less noticeable changes in nutrient content compared to shorter exposures. Previous studies have shown that nitrogen is lost in the form of gas due to high-temperature treatment and combustion of organic matter, leading to a decrease in

soil organic matter and total nitrogen content. This process also enhances the utilization efficiency of available potassium by crops [36]. In the current study, available potassium exhibited varying changes with different instantaneous high temperature treatments, but the overall average value increased. This suggests that short-term instantaneous high temperature heating can boost the availability of potassium in soil, thereby promoting better plant absorption. It can be seen that short-term high-temperature heating significantly impacts the physical, chemical, and biological properties of soil by decreasing soil pH, reducing soil organic matter and total nitrogen content, increasing microbial activity, and enhancing the availability of potassium in the soil [18, 27–32]. Furthermore, soil properties are influenced by both soil moisture and soil type. The soil moisture can impact the rate of heat conduction and organic matter decomposition, while various types of soil exhibit distinct responses to heat due to differences in mineral composition and structure [33]. As a result, these factors collaborate to alter the characteristics of the soil.

Comprehensive assessment of soil physical and chemical properties along with crop growth indicators under various high temperature treatments is crucial for enhancing soil quality and crop productivity [37, 38]. Principal component analysis proves to be more effective in extracting key factors influencing soil properties and crop growth when multiple evaluation criteria are considered [39]. In this research, six principal components were derived from 26 variables representing 19 indicators, capturing 86.291% of the original data without losing any information. These components can effectively substitute the original indicators for evaluation purposes. The study demonstrates the reliability of principal component analysis in evaluating soil properties and crop growth under different high temperature treatments. The comprehensive score of the CK treatment ranked 17th, indicating that 16 treatments outperformed the control in terms of soil quality. This suggests that high temperature treatments can enhance overall soil quality, but the specific temperature and duration of treatment can lead to varying outcomes. The top 3 treatments that yielded positive results in the comprehensive evaluation were 300°C for 6 seconds, 200°C for 14 seconds, and 400°C for 14 seconds. It is possible that the elevated temperatures in these treatments facilitated the breakdown of organic matter and the release of nutrients in the soil, ultimately benefiting the growth and development of crops. Additionally, these treatments might have suppressed the proliferation of pathogenic microorganisms and weeds in the soil, thereby enhancing the crop yield and optimizing the growing environment [15, 16, 31, 34]. Hence, it is essential to carefully select the appropriate temperature and duration for different treatment conditions to enhance the overall quality of both soil and crops.

The specific concept of this research involves subjecting the soil surface to high-temperature heating, which influences its physical and chemical properties through thermal conduction. Subsequently, corn is planted after the soil has been cooled down to examine the variations in corn growth resulting from enhanced soil properties. By briefly treating the soil temperature, it can promote the activity of soil microorganisms, enhance the nutrient supply capacity of the soil, improve soil quality, and increase both fertility and water retention. This can inspire researchers to explore high-temperature equipment for research and development purposes and subsequently facilitate its application in agriculture. This approach offers energy savings and improved efficiency compared to traditional temperature treatments. The study was conducted under controlled pot conditions, which differ from actual field cultivation of corn, posing challenges in generalizing findings to field environments with varying biological and abiotic conditions. Additionally, measurements were limited to a single planting cycle, lacking verification across multiple cycles. Future research will involve multi-cycle studies for result validation, utilizing simulation modeling to assess results across space and time. Moreover, the principal component analysis method utilized in the study has limitations, particularly in complex

biological and abiotic conditions. Subsequent studies may benefit from integrating other analytical methods for a more comprehensive understanding of the data.

## Conclusion

Different temperature and time treatments significantly influenced maize growth and soil physicochemical properties. Maize root growth, plant height, and leaf length were enhanced under specific treatment conditions, leading to changes in soil nutrient content and physical and chemical properties. Through principal component analysis and composite scores, a more comprehensive assessment of these effects can be conducted. The 300°C+6s treatment showed the best performance in the comprehensive evaluation, with the most significant impact on soil and crop quality. The findings of this study offer valuable insights for agricultural production and soil management, aiming to enhance crop growth, improve soil quality, and support sustainable agricultural development.

## Supporting information

**S1 Table. Growth index and root growth characteristics of maize.**
(XLSX)

**S2 Table. Physical and chemical properties of soil after crop harvest.**
(XLSX)

## Acknowledgments

The author thanks the Shaanxi Provincial Special Talent Support Program for its outstanding talent project research ideas and support for this study.

## Author Contributions

**Conceptualization:** Zhen Guo.

**Data curation:** Yang Zhang.

**Formal analysis:** Hua Zhuang.

**Funding acquisition:** Jichang Han.

**Investigation:** Jichang Han.

**Methodology:** Zhen Guo, Yang Zhang.

**Project administration:** Jichang Han.

**Resources:** Zhen Guo.

**Supervision:** Hua Zhuang.

**Writing – original draft:** Zhen Guo.

**Writing – review & editing:** Zhen Guo.

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
