## [Decision Letter · Decision Letter 0]

24 May 2024

PONE-D-24-13037

The combined effect of temperature on soil and maize was evaluated based on principal component analysisPLOS ONE

Dear Dr. Han,

Thank you for submitting your manuscript to PLOS ONE. After careful consideration, we feel that it has merit but does not fully meet PLOS ONE’s publication criteria as it currently stands. Therefore, we invite you to submit a revised version of the manuscript that addresses the points raised during the review process. Comments from the editorial office: One or more of the reviewers has recommended that you cite specific previously published works in your original submission PONE-D-24-07715 such as : 10.1080/15226514.2024.2310001; 10.1016/j.apsoil.2022.104680; 10.1016/j.scitotenv.2023.167756; 10.1016/j.scitotenv.2023.169420; 10.1016/j.stress.2023.100345; 10.1016/j.jhazmat.2021.127021 . Members of the editorial team have determined that the works referenced are not directly related to the submitted manuscript. As such, please note that it is not necessary or expected to cite the works requested by the reviewer and these may be removed from your submission if you wish.

We look forward to receiving your revised manuscript.

Kind regards,

Mojtaba Kordrostami, Ph.D.

Academic Editor

PLOS ONE

2. PLOS requires an ORCID iD for the corresponding author in Editorial Manager on papers submitted after December 6th, 2016. Please ensure that you have an ORCID iD and that it is validated in Editorial Manager. To do this, go to ‘Update my Information’ (in the upper left-hand corner of the main menu), and click on the Fetch/Validate link next to the ORCID field. This will take you to the ORCID site and allow you to create a new iD or authenticate a pre-existing iD in Editorial Manager. Please see the following video for instructions on linking an ORCID iD to your Editorial Manager account: https://www.youtube.com/watch?v=_xcclfuvtxQ.

Reviewers' comments:

Reviewer's Responses to Questions

**Comments to the Author**

1. Is the manuscript technically sound, and do the data support the conclusions?

Reviewer #1: Partly

Reviewer #2: Yes

2. Has the statistical analysis been performed appropriately and rigorously? 

Reviewer #1: Yes

Reviewer #2: Yes

3. Have the authors made all data underlying the findings in their manuscript fully available?

Reviewer #1: No

Reviewer #2: Yes

4. Is the manuscript presented in an intelligible fashion and written in standard English?

Reviewer #1: No

Reviewer #2: Yes

5. Review Comments to the Author

Reviewer #1: Kindly check the sentences and spell check like in line number 57-58 in introduction. It has been mentioned he high-temperature treatment is a commonly used method in soil facility management. It should be soil fertility not soil facility. Similarly in first line of introduction, it will be more appropriate to use soil fertility instead of soil infertility in line 41-42, The expansion of large-scale and intensive agriculture has worsened soil infertility due to

42 unsustainable farming practices"

As also earlier revisions, it was asked to explain whether rising soil temperature is s sustainable method of managing the soil pathogens. what was the effect of all these temperature treatments on soil beneficial microbes, soil enzymatic activities etc.

It is also not a usual impact of increase in soil temperature on root and shoot parameters.

The data to give information in support of defined objectives need better clarity. especially on The objectives of this study were to (1) investigate the effects of temperature treatments on soil properties over varying time scales; (2) clarify the response of maize growth indicators to temperature gradients; (3) clarify the optimal temperature under high-temperature treatments. We hypothesized that (1) the high temperature treatment can promote the comprehensive quality of soil and maize; (2) the short-term high temperature treatments would be more effective.

Reviewer #2: THE COMBINED EFFECT OF TEMPERATURE ON SOIL AND MAIZE WAS EVALUATED BASED ON PRINCIPAL COMPONENT ANALYSIS- PONE-D-24-13037

This study investigated the impact of temperature-modified soil on nutrient activation and crop growth, using loess soil with low fertility and maize as the test plant. Overall, the paper was well written, with clear objectives, detailed methodology, and systematic description of the results.

Overall Decision: Minor revision

TOPIC

1. Rephrase the topic. It currently reads like the objective of the study

2. Arrange key words in alphabetical order

ABSTRACT & KEYWORDS

1. Lines 14-16: The problem statement is not clearly defined

INTRODUCTION

1. Line 79-80. Using the principal component analysis method to evaluate soil physicochemical properties and crop growth indexes. The sentence is hanging. Rephrase.

2. Overall, the Introduction section is well written

MATERIALS AND METHODS

1. Lines 89-98: What is general classification of the soil with reference to common nomenclature e.g. WRBS (FAO), USDA

2. Lines 89-98: Rationale for site selection should be further strengthened. What does the Loess Plateau represent in the broader crop production systems in China and similar farming systems in the world?

3. Lines 101-113: What informed/guided the temperature and heating periods used in the study

4. It’s not clearly stated in the Materials and Methods the time period (duration of the experiment) the experiment was done. In addition, it is important to provide information of the days to maturity of the maize variety used in this study

5. Was the soil characterized for physicochemical and biological properties before experimentation?

RESULTS & DISCUSSION

1. Tables 1 and 2: Were the treatments significantly different? There no statistical indices for comparison of treatment means.

2. Limitations of the study and grey areas for further research should be clearly articulated in the Discussion.

• The study was conducted under controlled (pot experiment) conditions, which does not represent the real field conditions where farmers grow maize. It therefore becomes challenging to extrapolate the findings to field conditions where both biotic and abiotic conditions vary in space and time.

• Measurements were only done over one cropping cycle. The study could have been conducted over 2 or more cropping cycles to validate the results. In the absence of these repeated measurements over 2 or more seasons, the authors could consider simulation modeling to validate their findings in space and time

• Limitations of Principal Component Analysis in the context of this study

3. I was wondering, what are the implications of the study findings regarding management of temperature regimes in soils and impacts on soil chemical and biological properties in maize-based cropping systems. For example, how can farmers realistically manipulate soil temperatures to influence soil chemical and biological properties and crop growth?

6. PLOS authors have the option to publish the peer review history of their article (what does this mean?). If published, this will include your full peer review and any attached files.

Reviewer #1: **Yes: **Sanjay Singh Rathore

Reviewer #2: No

---

## [Author Response · Author response to Decision Letter 0]

30 May 2024

PONE-D-24-13037

The combined effect of temperature on soil and maize was evaluated based on principal component analysis

PLOS ONE

Dear Dr. Han,

Thank you for submitting your manuscript to PLOS ONE. After careful consideration, we feel that it has merit but does not fully meet PLOS ONE’s publication criteria as it currently stands. Therefore, we invite you to submit a revised version of the manuscript that addresses the points raised during the review process.

Comments from the editorial office: One or more of the reviewers has recommended that you cite specific previously published works in your original submission PONE-D-24-07715 such as : 10.1080/15226514.2024.2310001; 10.1016/j.apsoil.2022.104680; 10.1016/j.scitotenv.2023.167756; 10.1016/j.scitotenv.2023.169420; 10.1016/j.stress.2023.100345; 10.1016/j.jhazmat.2021.127021 . Members of the editorial team have determined that the works referenced are not directly related to the submitted manuscript. As such, please note that it is not necessary or expected to cite the works requested by the reviewer and these may be removed from your submission if you wish.

Response: The author carefully revised the references based on the editor's feedback.

References 1, 5, 10, 27, and 30, which are closely related to the content of the article, were cited.

Response: In the materials and methods section, the author provided a detailed description of the experimental design, operation procedure, and detection method related to the experimental protocols in this article.

Response: Our manuscript meets the stylistic requirements of PLOS ONE.

2. PLOS requires an ORCID iD for the corresponding author in Editorial Manager on papers submitted after December 6th, 2016. Please ensure that you have an ORCID iD and that it is validated in Editorial Manager. To do this, go to ‘Update my Information’ (in the upper left-hand corner of the main menu), and click on the Fetch/Validate link next to the ORCID field. This will take you to the ORCID site and allow you to create a new iD or authenticate a pre-existing iD in Editorial Manager. Please see the following video for instructions on linking an ORCID iD to your Editorial Manager account: https://www.youtube.com/watch?v=_xcclfuvtxQ.

Response: The author has provided ORCID iD: 0000-0002-1609-9848

Reviewer #1

 Kindly check the sentences and spell check like in line number 57-58 in introduction. It has been mentioned he high-temperature treatment is a commonly used method in soil facility management. It should be soil fertility not soil facility. 

Response: The author has revised to replace "soil facility" with "soil fertility".

Similarly in first line of introduction, it will be more appropriate to use soil fertility instead of soil infertility in line 41-42, The expansion of large-scale and intensive agriculture has worsened soil infertility due to unsustainable farming practices"

Response: The author has revised the sentence “The expansion of large-scale and intensive agriculture has negatively impacted soil fertility through unsustainable farming practices".

As also earlier revisions, it was asked to explain whether rising soil temperature is s sustainable method of managing the soil pathogens. what was the effect of all these temperature treatments on soil beneficial microbes, soil enzymatic activities etc.

Response: The last paragraph of the introduction has been rewritten by the author to answer the questions raised by the reviewers.

“Soil temperature and heating duration are key factors affecting the physicochemical characteristics of soil and crop development. The short-term exposure to high temperatures can quickly enhance the soil conditions, proving to be a more effective approach than prolonged high temperature maintenance throughout the entire growth cycle. The short high temperature treatment can stimulate soil microbial activity, support the proliferation and function of beneficial microorganisms, and increase the activity of enzymes, thereby accelerating the decomposition of organic matter”. 

It is also not a usual impact of increase in soil temperature on root and shoot parameters.

The data to give information in support of defined objectives need better clarity. especially on The objectives of this study were to (1) investigate the effects of temperature treatments on soil properties over varying time scales; (2) clarify the response of maize growth indicators to temperature gradients; (3) clarify the optimal temperature under high-temperature treatments. We hypothesized that (1) the high temperature treatment can promote the comprehensive quality of soil and maize; (2) the short-term high temperature treatments would be more effective.

Response: In this study, the soil surface underwent high temperature treatment, impacting the activities of microorganisms and enzymes, consequently affecting soil properties. Subsequently, maize was planted in the cooled soil treated with high temperature to observe growth differences, representing the innovation of this research. As a result, the author adjusted the study's purpose to highlight these distinctions.

“The objectives of this study were to (1) investigate the impact of short-term high-temperature treatment on soil properties; (2) explore the difference between planting maize in soil treated with high temperatures; (3) identify the optimal combination of temperature and duration. We hypothesized that (1) the high temperature treatment can promote the comprehensive quality of soil and maize; (2) the short-term high temperature treatments would be more effective”.

Reviewer #2

THE COMBINED EFFECT OF TEMPERATURE ON SOIL AND MAIZE WAS EVALUATED BASED ON PRINCIPAL COMPONENT ANALYSIS- PONE-D-24-13037

This study investigated the impact of temperature-modified soil on nutrient activation and crop growth, using loess soil with low fertility and maize as the test plant. Overall, the paper was well written, with clear objectives, detailed methodology, and systematic description of the results.

Overall Decision: Minor revision

Response: Thank you for your comments, I will modify and improve according to your comments.

TOPIC

1. Rephrase the topic. It currently reads like the objective of the study

2. Arrange key words in alphabetical order

Response: 1. The author revised the title according to the opinion. Title: The impact of high-temperature treatments on maize growth parameters and soil nutrients: A comprehensive evaluation through principal component analysis

2. Keywords are listed in alphabetical order.

ABSTRACT & KEYWORDS

1. Lines 14-16: The problem statement is not clearly defined

Response: The author redefines the research question.

Line 14-15: It is important to investigate the impact of short-term high temperature exposure on maize growth and rhizosphere soil nutrients, in comparison to long-term high temperature treatment.

INTRODUCTION

1. Line 79-80. Using the principal component analysis method to evaluate soil physicochemical properties and crop growth indexes. The sentence is hanging. Rephrase.

2. Overall, the Introduction section is well written

Response: The author has revised the sentence “The soil physicochemical properties and crop growth indexes are evaluated using the principal component analysis method”.

MATERIALS AND METHODS

1. Lines 89-98: What is general classification of the soil with reference to common nomenclature e.g. WRBS (FAO), USDA

Response: Soil classification using Chinese soil system classification principles, 2009 edition. The test soil is loessal soil (China Soil System Classification 2009, CAS).

2. Lines 89-98: Rationale for site selection should be further strengthened. What does the Loess Plateau represent in the broader crop production systems in China and similar farming systems in the world?

Response: In China's crop production system, the Loess Plateau exemplifies an agricultural production model that is tailored to arid and semi-arid climate conditions. Globally, the agricultural system of the Loess Plateau serves as a model for agricultural development in arid and semi-arid regions, offering valuable insights for similar areas.

3. Lines 101-113: What informed/guided the temperature and heating periods used in the study

Response: The author has added in the experimental design. 

The temperature and heating time used in the experiment were determined by relevant research results as well as previous experimental studies and research assumptions. 

4. It’s not clearly stated in the Materials and Methods the time period (duration of the experiment) the experiment was done. In addition, it is important to provide information of the days to maturity of the maize variety used in this study

Response: The author has added in the Materials and Methods. 

The duration of the experiment was 95 days during the whole growth period of maize.

After cooling, maize seeds (Zhengdan 958) were sown in each pot using a drilling method

During the maize growth period, which lasts 95 days, measurements were taken for crop roots, plant height, and leaf length. 

5. Was the soil characterized for physicochemical and biological properties before experimentation?

Response: The predominant soil type in the research area is loessal soil, characterized by a pH of 8.04, SOM content of 14.25 g kg-1, TN content of 0.58 g kg-1, AP content of 19.06 mg kg-1, AK content of 175.75 mg kg-1, and a silty loam texture.

RESULTS & DISCUSSION

1. Tables 1 and 2: Were the treatments significantly different? There no statistical indices for comparison of treatment means.

Response: The authors present the values of these indicators in a table. On this basis, a series of tests and evaluations were made using principal component analysis, and the final conclusion was obtained by the principal component statistical analysis method, which was used to determine the significant differences between different treatments. We will also take care to ensure that data and statistical analysis results are presented more clearly in future writing to improve the readability and scientific nature of the paper. Thank you.

2. Limitations of the study and grey areas for further research should be clearly articulated in the Discussion.

• The study was conducted under controlled (pot experiment) conditions, which does not represent the real field conditions where farmers grow maize. It therefore becomes challenging to extrapolate the findings to field conditions where both biotic and abiotic conditions vary in space and time.

• Measurements were only done over one cropping cycle. The study could have been conducted over 2 or more cropping cycles to validate the results. In the absence of these repeated measurements over 2 or more seasons, the authors could consider simulation modeling to validate their findings in space and time

• Limitations of Principal Component Analysis in the context of this study

Response: The authors add limitations to the study at the end of the discussion.

The study was conducted under controlled pot conditions, which differ from actual field cultivation of corn, posing challenges in generalizing findings to field environments with varying biological and abiotic conditions. Additionally, measurements were limited to a single planting cycle, lacking verification across multiple cycles. Future research will involve multi-cycle studies for result validation, utilizing simulation modeling to assess results across space and time. Moreover, the principal component analysis method utilized in the study has limitations, particularly in complex biological and abiotic conditions. Subsequent studies may benefit from integrating other analytical methods for a more comprehensive understanding of the data.

3. I was wondering, what are the implications of the study findings regarding management of temperature regimes in soils and impacts on soil chemical and biological properties in maize-based cropping systems. For example, how can farmers realistically manipulate soil temperatures to influence soil chemical and biological properties and crop growth?

Response: The specific idea of this research involves heating the surface of the soil at high temperatures, which affects its physical and chemical properties through temperature conduction. Subsequently, corn is planted after the soil is cooled to investigate the differences in corn growth following the improvement of soil properties. This study aims to offer a significant theoretical foundation and practical value for agricultural production.

 The results help to understand the effects of different temperatures on soil chemistry and biological properties, thereby providing farmers with strategies to improve soil quality. Through short soil temperature treatment, the activity of soil microorganisms can be promoted, the nutrient supply capacity of soil can be improved, the soil quality can be improved, and the soil fertility and water retention can be increased.

It can stimulate researchers' research and development and exploration of high-temperature equipment, and then promote it to agricultural use, which can save energy and increase efficiency compared with traditional temperature treatment.

---

## [Decision Letter · Decision Letter 1]

17 Jul 2024

PONE-D-24-13037R1The impact of high-temperature treatments on maize growth parameters and soil nutrients: A comprehensive evaluation through principal component analysisPLOS ONE

Dear Dr. Han,

Thank you for submitting your manuscript to PLOS ONE. After careful consideration, we feel that it has merit but does not fully meet PLOS ONE’s publication criteria as it currently stands. Therefore, we invite you to submit a revised version of the manuscript that addresses the points raised during the review process.

We look forward to receiving your revised manuscript.

Kind regards,

Mojtaba Kordrostami, Ph.D.

Academic Editor

PLOS ONE

Journal Requirements:

Reviewers' comments:

Reviewer's Responses to Questions

**Comments to the Author**

1. If the authors have adequately addressed your comments raised in a previous round of review and you feel that this manuscript is now acceptable for publication, you may indicate that here to bypass the “Comments to the Author” section, enter your conflict of interest statement in the “Confidential to Editor” section, and submit your "Accept" recommendation.

Reviewer #1: (No Response)

Reviewer #2: All comments have been addressed

2. Is the manuscript technically sound, and do the data support the conclusions?

Reviewer #1: Partly

Reviewer #2: Yes

3. Has the statistical analysis been performed appropriately and rigorously? 

Reviewer #1: N/A

Reviewer #2: Yes

4. Have the authors made all data underlying the findings in their manuscript fully available?

Reviewer #1: Yes

Reviewer #2: Yes

5. Is the manuscript presented in an intelligible fashion and written in standard English?

Reviewer #1: Yes

Reviewer #2: Yes

6. Review Comments to the Author

Reviewer #1: It seems authors couldn’t understand point raised like on soil texure, authors explain it was change in soil particles. But texture is different soil property. Authors once again requested to go the the comments carefully and give proper reasoning. This must also be reflected in revised MS. Superficial reply of the issues raised must be avoided.

Reviewer #2: This study investigated the impact of temperature-modified soil on nutrient activation and crop growth, using loess soil with low fertility and maize as the test plant. Overall, the paper was well written, with clear objectives, detailed methodology, and systematic description of the results.

Overall Decision: Accept

ABSTRACT & KEYWORDS

1. The problem statement is not clearly defined

RESULTS & DISCUSSION

1. I was wondering, what are the implications of the study findings regarding management of temperature regimes in soils and impacts on soil chemical and biological properties in maize-based cropping systems. For example, how can farmers realistically manipulate soil temperatures to influence soil chemical and biological properties and crop growth?

The authors gave a satisfactory response under “responses to reviewers’ comments” document but did not include the explanation in the actual Discussion section. Add the response to the Discussion section.

7. PLOS authors have the option to publish the peer review history of their article (what does this mean?). If published, this will include your full peer review and any attached files.

Reviewer #1: **Yes: **Sanjay Singh Rathore

Reviewer #2: No

---

## [Author Response · Author response to Decision Letter 1]

25 Jul 2024

PONE-D-24-13037R1

The impact of high-temperature treatments on maize growth parameters and soil nutrients: A comprehensive evaluation through principal component analysis

PLOS ONE

Reviewer #1: It seems authors couldn’t understand point raised like on soil texure, authors explain it was change in soil particles. But texture is different soil property. Authors once again requested to go the the comments carefully and give proper reasoning. This must also be reflected in revised MS. Superficial reply of the issues raised must be avoided.

Response: I am very sorry that the author misunderstood the definition of soil texture. The author fully accepts the reviewer's definition of soil texture, which is different soil properties.

The author reanswers the question of how the texture of the soil is affected. Add in the second paragraph of the discussion. The specific contents are as follows:

It can be seen that short-term high-temperature heating significantly impacts the physical, chemical, and biological properties of soil by decreasing soil pH, reducing soil organic matter and total nitrogen content, increasing microbial activity, and enhancing the availability of potassium in the soil. Furthermore, soil properties are influenced by both soil moisture and soil type. The soil moisture can impact the rate of heat conduction and organic matter decomposition, while various types of soil exhibit distinct responses to heat due to differences in mineral composition and structure. As a result, these factors collaborate to alter the characteristics of the soil.

Reviewer #2: This study investigated the impact of temperature-modified soil on nutrient activation and crop growth, using loess soil with low fertility and maize as the test plant. Overall, the paper was well written, with clear objectives, detailed methodology, and systematic description of the results.

Overall Decision: Accept

ABSTRACT & KEYWORDS

1. The problem statement is not clearly defined

Response: The author has revised it, and the details are as follows:

In contrast to prolonged exposure to high temperatures, investigating short-term high-temperature stress can provide insights into the impact of varying heat stress durations on plant development and soil nutrient dynamics, which is crucial for advancing ecological agriculture. 

RESULTS & DISCUSSION

1. I was wondering, what are the implications of the study findings regarding management of temperature regimes in soils and impacts on soil chemical and biological properties in maize-based cropping systems. For example, how can farmers realistically manipulate soil temperatures to influence soil chemical and biological properties and crop growth?

The authors gave a satisfactory response under “responses to reviewers’ comments” document but did not include the explanation in the actual Discussion section. Add the response to the Discussion section.

Response: The author has added specific replies to the last paragraph of the discussion section, further enhancing the practicality of the technique. The specific contents are as follows:

The specific concept of this research involves subjecting the soil surface to high-temperature heating, which influences its physical and chemical properties through thermal conduction. Subsequently, corn is planted after the soil has been cooled down to examine the variations in corn growth resulting from enhanced soil properties. By briefly treating the soil temperature, it can promote the activity of soil microorganisms, enhance the nutrient supply capacity of the soil, improve soil quality, and increase both fertility and water retention. This can inspire researchers to explore high-temperature equipment for research and development purposes and subsequently facilitate its application in agriculture. This approach offers energy savings and improved efficiency compared to traditional temperature treatments.

---

## [Editor Report · Decision Letter 2]

6 Aug 2024

The impact of high-temperature treatments on maize growth parameters and soil nutrients: A comprehensive evaluation through principal component analysis

PONE-D-24-13037R2

Dear Dr. Han,

We’re pleased to inform you that your manuscript has been judged scientifically suitable for publication and will be formally accepted for publication once it meets all outstanding technical requirements.

Kind regards,

Mojtaba Kordrostami, Ph.D.

Academic Editor

PLOS ONE
---

## [Editor Report · Acceptance letter]

7 Aug 2024

PONE-D-24-13037R2 

PLOS ONE

Dear Dr. Han, 

I'm pleased to inform you that your manuscript has been deemed suitable for publication in PLOS ONE. Congratulations! Your manuscript is now being handed over to our production team.

Kind regards, 

on behalf of

Dr. Mojtaba Kordrostami 

Academic Editor

PLOS ONE